# Catalytic Oxidation of Acetone over MnO_x_-SiO_2_ Catalysts: An Effective Approach to Valorize Rice Husk Waste

**DOI:** 10.3390/ma17246069

**Published:** 2024-12-12

**Authors:** Mauricio Cardoso, Patrice Portugau, Carolina De Los Santos, Ricardo Faccio, Hilario Vidal, José Manuel Gatica, María del Pilar Yesté, Jorge Castiglioni, Martin Torres

**Affiliations:** 1Área Fisicoquímica, Facultad de Química, Universidad de la República, Gral. Flores 2124, Montevideo 11800, Uruguay; 2Área Física, Facultad de Química, Universidad de la República, Gral. Flores 2124, Montevideo 11800, Uruguay; 3Departamento C.M. I.M. y Química Inorgánica, Universidad de Cádiz, 11510 Puerto Real, Spain

**Keywords:** rice husk silica, biomass waste, VOC, manganese oxides, catalyst, air pollution

## Abstract

Rice husk, a byproduct of rice production, poses significant environmental challenges due to disposal issues, while the emission of volatile organic compounds into the atmosphere further exacerbates these concerns. This study addresses both problems by exploring the potential of texturally enhanced SiO_2_, derived from Uruguayan rice husk, as a catalytic support for manganese oxides in the combustion of volatile organic compounds. SiO_2_ was synthesized from rice husk ash using a sustainable, acid-free pretreatment method, yielding a notably high silica purity of 96.5%—a level comparable to or exceeding previously reported values, highlighting the high silica quality inherent in Uruguayan rice husk. The catalytic activity was evaluated using acetone as a model volatile organic compound, achieving up to 90% conversion with 30 wt.% manganese oxide at 300 °C, with CO_2_ as the primary product. Furthermore, a 24 h stability test demonstrated consistent performance, maintaining a conversion rate of around 95.6 ± 2.5%. These findings suggest that high-purity SiO_2_ derived from Uruguayan rice husk, with its sustainability benefits, offers an effective solution for acetone removal when supporting an active phase such as manganese oxides, addressing both rice husk disposal and volatile organic compound emissions.

## 1. Introduction

The conversion of biomass waste globally to produce value-added materials is a current trend, driven by the United Nations Sustainable Development Goals. One of the main objectives is the modernization of infrastructure and the transition of industries towards sustainability through more efficient use of resources and the promotion of clean and environmentally responsible technologies and industrial processes. This implies that all countries must take measures according to their respective capacities [1].

In this sense, Uruguay is renowned for being an agrarian country that extensively utilizes its soil for various crops, particularly rice. During the 2022–2023 season, the estimated rice harvest reached 1,462,800 tonnes, as reported by the competent ministry of Uruguay [2]. This level of production places Uruguay among the top 10 rice producers [3]. The primary product of the rice crop is paddy rice, accounting for 80% of the total harvested mass [4], while the remaining 20% corresponds to rice husk (RH), which is predominantly used for energy production due to its high calorific value of 14.7 MJ kg^−1^ [5]. In Uruguay, there are two energy production plants that utilize RH as a fuel source [6]. Despite its extensive use as an energy source, a significant portion of the rice husk produced remains unused, ending up deposited in various locations, including watercourses, causing environmental degradation [4]. This situation raises concerns among the Uruguayan academic community, which has been exploring multiple approaches for RH management and valorization, including its use in energy storage and the production of antioxidants, among others [7,8,9].

When RH is used to produce energy, the challenge lies in managing large quantities of ash, which is primarily composed of amorphous SiO_2_ (90–95%) [4,10]. These ashes are not widely utilized and are eventually deposited in the environment, causing soil acidification and alterations in its physicochemical properties [11]. The abundance of SiO_2_ in RH ash has prompted its evaluation as a potential material for various applications [4].

The SiO_2_ obtained from RH combustion at temperatures below 800 °C is mainly amorphous but requires textural modification for broader applications in material science. These modifications are achieved via specific reaction sequences that yield high specific surface areas [12]. Texturally enhanced SiO_2_ derived from RH has multiple applications, including adsorption, supercapacitors, controlled release systems, and catalysis, among others [4,10,13,14,15].

SiO_2_ has been evaluated as a catalyst support by incorporating active species. This application has been widely reported in studies spanning a variety of reactions, such as water contaminant remediation, the Fischer–Tropsch reaction, hydrogenation reactions, and the combustion of volatile organic compounds (VOCs). In the latter case, active phases such as copper and palladium have been utilized [16,17,18,19].

Currently, atmospheric pollution is a significant concern for the scientific community. The emission of VOCs into the atmosphere resulting from industrial and anthropogenic activities has numerous environmental and human health impacts due to their toxicity and contribution to the formation of ground-level ozone and photochemical smog [20,21]. Catalytic combustion has emerged as a promising solution to prevent the emission of VOCs into the atmosphere [21]. In this context, manganese-based catalysts have demonstrated broad efficacy for the conversion of VOCs [22].

Manganese is one of the most abundant metals in the Earth’s crust, resulting in the availability of cost-effective manganese compounds. This transition metal features five unpaired d-electrons, enabling it to exhibit multiple oxidation states and form a wide range of stable oxides (MnO_x_) [23]. Moreover, manganese is capable of transitioning between oxidation states, such as MnO_2_, Mn_2_O_3_, and Mn_3_O_4_ [24]. This capability explains its excellent performance in VOC oxidation. Despite its catalytic combustion capabilities, treating gas streams containing VOCs requires large quantities of MnO_x_ due to its low specific surface area (<5 m^2^ g^−1^). Consequently, supporting MnO_x_ on a mesoporous solid enhances dispersion and increases atomic efficiency, offering an effective method to boost the specific surface area and improve catalytic performance [25].

Our research group has assessed the purity of RH ash in terms of SiO_2_ content, obtained from Uruguayan RH, and found it to exceed 95% without necessitating further purification steps. These results establish Uruguayan RH as a high-purity silica source, differentiating it from other studies that require acid treatments to achieve similar purity levels [12,26]. Based on the aforementioned information, and to the best of the authors’ knowledge, this is the first time RH ash has been evaluated from this perspective. The objective of this study is to develop a new type of catalyst, characterize the texturally enhanced SiO_2_ derived from Uruguayan rice husk ash, and assess its potential as a catalytic support for MnO_x_ in the catalytic combustion of a VOC such as acetone. This represents a new application for rice husk waste in the field of catalysis [27]. Furthermore, this study provides insights into a new method for valorizing rice husk ash, offering a foundation for potential solutions to two pressing environmental challenges.

## 2. Materials and Methods

### 2.1. Synthesis of SiO_2_

Rice husk (RH) from the *Olimar* variety, supplied by the Sociedad Anónima de Molinos Arroceros Nacionales (SAMAN, Montevideo, Uruguay), was used as the raw material. RH underwent a pre-treatment process: a large quantity was ground and sieved to collect 50 g of material with a particle size between 0.60 and 0.80 mm. The sieved sample was washed with distilled water to remove soil and other soluble impurities and then dried for 24 h at 105 °C.

The synthesis of SiO_2_ from RH followed the technique described in [12] with some modifications. The pre-treated RH was directly calcined at 600 °C for 2 h in a muffle furnace, yielding approximately 10 g of ash. After calcination, the resulting RH ash was dissolved in NaOH (1 g of RH ash per 10 mL of 4 M NaOH) with continuous magnetic stirring and maintained at 80 °C for 4 h, as shown in Equation (1).
(1)$$SiO_{2}\;\left(s\right)+2NaOH\;\left(aq\right)\overset{\Delta }{\to }\;Na_{2}SiO_{3}\;\left(aq\right)+H_{2}O\;(l)$$

The solution was cooled to room temperature and filtered. The pH was measured, and the solution was stored in a polypropylene flask. It was then treated with a stoichiometric volume of 5 M HCl added in one step to achieve a pH of 9. At this pH, silicic acid gelled within the solution, as illustrated in Equation (2).
(2)$$Na_{2}SiO_{3}\;\left(aq\right)+2HCl\;\to 2NaCl\;\left(aq\right)+H_{2}SiO_{3}\;(s)$$

The gel was washed multiple times under magnetic stirring conditions to ensure mass exchange and reduce impurities until a neutral pH and a conductivity below 50 µS cm^−1^ were reached. The gel was then dried overnight at 105 °C in an oven. Finally, the solid product was calcined in an air atmosphere at 600 °C for 2 h to obtain SiO_2_, following the process shown in Equation (3). This final solid was labeled “RHS”.
(3)$$H_{2}SiO_{3}\;\left(s\right)\overset{\Delta }{\to }\;SiO_{2}\;\left(s\right)+H_{2}O\;(l)$$

### 2.2. Synthesis of Catalysts

Catalysts were synthesized via the incipient wetness impregnation method [24,28], using manganese (II) acetate tetrahydrate (Mn(Ac)_2_), supplied by Sigma-Aldrich (St. Louis, MO, USA) (purity > 99%), as the MnO_x_ precursor. Figure 1 summarizes the synthesis route followed to obtain the catalysts. An appropriate amount of Mn(Ac)_2_ was dissolved in distilled water to create a saturated solution. The RHS was impregnated with this solution, then dried in an oven at 105 °C for 24 h.

The impregnated solid was calcined in a furnace with the following temperature pro-gramme: heating from room temperature to 550 °C at 5 °C min^−1^, followed by a 4 h dwell at 550 °C. This program was based on thermal analysis results (discussed below). Catalysts were labeled as MnX/RHS, where X denotes the weight percentage of manganese loaded onto RHS (10%, 30%, or 40%). Bulk manganese oxide (MnO_x_) was synthesized using the same precursor and temperature program.

### 2.3. Characterization Studies

#### 2.3.1. X-Ray Diffraction (XRD) and Rietveld Analysis

X-ray powder diffraction was performed using a Rigaku Miniflex 600-C (Tokyo, Japan) X-ray diffractometer in θ–θ geometry. The instrument was operated with a CuKα X-ray radiation source (sealed tube at 40 kV/30 mA) and a curved Ge crystal beam monochromator with a focal distance of 285 mm. Data were collected over a 2θ range of 2.00°–70.00° in steps of 0.02°, with a counting time of 2 s per step. Full-pattern profile fitting via Rietveld refinement [29] was conducted using Smart Lab Studio II software (version 4.3) from Rigaku (Tokyo, Japan).

#### 2.3.2. Thermogravimetric Analysis (TGA)

TGA was performed using a Shimadzu TGA-50 apparatus (Kyoto, Japan). Approximately 10 mg of each sample was analyzed in a platinum pan under an air stream (50 cm^3^ min^−1^). For RH, the temperature program ranged from room temperature to 900 °C at a heating rate of 10 °C min^−1^. For the MnO_x_ precursor, the temperature program ranged from room temperature to 550 °C at 5 °C min^−1^, followed by a 4 h dwell at 550 °C.

#### 2.3.3. Microwave-Induced Plasma Atomic Emission Spectrometry (MIP-AES)

The purity of RHS ash was assessed using an Agilent 4210 MIP-OES system (Santa Clara, CA, USA). For analysis, 250 mg of sample was dissolved in 15 mL of 0.5 M KOH, sonicated for 20 min, and treated in a water bath at 80 °C for 3 h.

#### 2.3.4. Textural Characterization

Textural properties were determined via N_2_ physisorption at 77 K using a Micromeritics ASAP2020 analyzer (Norcross, GA, USA). Samples were pre-evacuated at 200 °C for 2 h. The specific surface area was calculated using the BET method, and total pore volume was estimated from nitrogen adsorption at a relative pressure of 0.99. The Non-Local Density Functional Theory (NLDFT) method was applied to determine the pore size distribution.

#### 2.3.5. Fourier Transformed Infrared Spectroscopy (FTIR)

FTIR spectra were obtained using a Shimadzu QATR-S IRSpirit spectrometer (Kyoto, Japan) in the range of 4000–400 cm^−1^. Measurements were recorded in % Transmittance mode with attenuated total reflection (ATR) using a single-reflection accessory (QATR-S).

#### 2.3.6. Scanning Electron Microscopy (SEM) and Energy Dispersive Spectroscopy (EDS)

SEM analysis was conducted on uncoated samples using a JEOL JS M-5900LV (Tokyo, Japan) scanning electron microscope operating at 25 kV. Samples were mounted on aluminum holders with carbon tape. Elemental composition was analyzed using an energy-dispersive spectrometry probe (NORAN System 7) at 20 kV.

#### 2.3.7. Temperature-Programmed Reduction Mass Spectrometry (H2-TPR)

H_2_-TPR experiments were carried out using a Micromeritics Autochem (Norcross, GA, USA) apparatus equipped with a thermal conductivity detector (TCD). Samples were pre-treated in Ar at 500 °C for 1 h (60 cm^3^ min^−1^) before cooling to 25 °C. During TPR, samples were heated to 900 °C (10 °C min^−1^) under 5% H₂/Ar (60 cm^3^ min^−1^) conditions for 1 h.

### 2.4. Catalytic Evaluation 

The catalytic activity of the materials for acetone oxidation was evaluated using 500 mg samples placed in a U-shaped quartz reactor. Oxygen was bubbled through liquid acetone at (0 ± 1) °C with a flow rate of 6.5 cm^3^ min^−1^, then mixed with an argon flow of 120 cm^3^ min^−1^. The gas hourly space velocity (GHSV) was approximately 15,000 cm^3^ h^−1^ g^−1^ cat.

For each test, the temperature was increased from 150 °C to 500 °C in increments of 50 °C. Gas samples were collected from the reactor inlet and outlet to monitor gas composition using a Shimadzu GC-2014 (Kyoto, Japan) gas chromatograph equipped with a Hayesep^®^ column. The stability of the most active catalyst was tested under the same flow conditions for 24 h at the temperatures identified from *light-off* curves.

## 3. Results and Discussion

### 3.1. Characterization Results

In the thermogravimetric analysis depicted in Figure 1a, RH exhibits three distinct mass loss steps: the first is attributed to moisture in the raw sample, while the subsequent two result from organic matter decomposition, consistent with the literature [8]. Above 500 °C, the mass stabilizes at approximately 20%, which is attributed to ash content. Figure 1b presents the thermal analysis for RHS and Mn(Ac)_2_, noting that RHS exhibits no weight loss above 105 °C, with mass loss observed only from humidity below 100 °C. The decomposition of Mn(Ac)_2_ to form MnO_x_ aligns with previous studies [24]. Consequently, the calcination temperature for the catalysts was set at 550 °C.

The SiO_2_ content in RHS, determined using MIP-AES, was found to be 96.5 wt.%, confirming the high purity of silica in our rice husk sample. Additionally, the sodium content, quantified as Na_2_O, was found to be 2.47 wt.%. Compared to previous studies that employed acid-based extraction methods [12,26], this approach presents a more sustainable alternative due to its reduced energy requirements. The acid-free extraction method not only enhances sustainability by minimizing chemical usage and energy consumption but also reinforces the purity of our RHS, highlighting its potential as a cost-effective and environmentally responsible source of high-purity silica for various applications.

Figure 2 shows the RHS diffractogram, which presents the typical broad dispersion of amorphous SiO_2_ in the 2θ range of approximately 20–30° [30]. Conversely, the bulk MnO_x_ diffractogram presents a diffraction pattern consistent with the crystalline structure of α-Mn_2_O_3_ (Bixbyite, cubic crystalline system, space group $Ia\bar{3}$) [24], as shown in Appendix A. According to a full-pattern profile fitting and Rietveld refinement [29,31] (see Appendix A), the refined crystalline size domain value corresponds to D = 26.8(4) nm, indicating a nanosized domain.

The synthesized catalysts (Mn10RHS, Mn30RHS, and Mn40RHS) exhibit a similar crystalline structure, and their broad reflections are related to the presence of amorphous SiO_2_, which indicates the structural stability of the support. For the supported catalysts, the diffraction pattern matches the crystalline structure of majority Mn_3_O_4_ (Hausmannite, tetragonal crystalline system, space group I4_1_/amd) and minority β-MnO_2_ (Pyrolusite, tetragonal crystalline system, space group P4_2_/mnm) [32]. The corresponding Bragg reflections display significant full widths at half maximum (FWHM), suggesting a possible confinement effect according to Scherrer’s equation [33]. Due to this broadening and the proximity of the Bragg reflections between Mn_3_O_4_ and β-MnO_2_, the samples are likely to contain both phases.

We applied an internal standard procedure using yttrium oxide (cubic crystalline system, space group $Ia\bar{3}$) as a reference pattern to determine the relative weight fractions of the crystalline and amorphous components. For Mn40RHS, we obtained a crystalline size domain value of approximately D = 30.9(5) nm and relative weight fractions on a crystalline basis of 17.44(2)% for Mn_3_O_4_, 0.04(2)% for β-MnO_2_, and 82.5(2)% for the amorphous phase. For Mn30RHS, we refined a crystalline size domain of D = 18.4(5) nm, with relative weight fractions of 14.4(4)% for Mn_3_O_4_, 1.1(4)% for β-MnO_2_, and 84.5% for the amorphous phase. Finally, for Mn10RHS, we obtained D = 10(1) nm for Mn_3_O_4_ exclusively, with a relative weight fraction of 5.7(4)%, accompanied by an amorphous fraction of 94.3%.

The difference between the crystalline structure of bulk MnO_x_ and that of the supported catalysts reveals the interaction of the SiO₂ surface during the formation of MnO_x_. Additionally, there are differences in the evolution of the crystalline unit cell parameters for the majority phase Mn_3_O_4_, with a = 5.7646(3) Å, a = 5.7737(16) Å, and a = 5.784(9) Å for Mn40RHS, Mn30RHS, and Mn10RHS, respectively. As mentioned above, the reducibility of Mn_2_O_3_ differs from that of Mn_3_O_4_; therefore, its catalytic performance is expected to vary accordingly.

The N₂ physisorption isotherm presented in Figure 3a, along with the corresponding pore size distribution (PSD) in Figure 3b, illustrates the characteristics of all catalysts. According to the IUPAC classification [34], the isotherms are primarily Type II with H3 hysteresis loops. The PSD profiles for all catalysts and the RHS support are similar; however, the pores measuring approximately 1 nm decrease as the manganese phase content increases, resulting in a reduction in the microporosity of the samples.

Table 1 lists the textural parameters and specific surface area (S_BET_) for the catalysts, with RHS exhibiting a S_BET_ of 333 m^2^/g, indicating a favorable surface area compared to other silica supports derived from RH ash [12]. For supported catalysts, S_BET_ decreases with the increase in MnO_x_ loading, a trend also observed in total pore volume (V_Tp_). This behavior likely results from the reduction in mesoporous volume (V_mp_) and micropore volume (V_µp_) due to catalyst deposition and subsequent pore coverage.

The FT-IR analysis of the synthesized catalysts and RHS, as shown in Figure 4, reveals that RHS displays three main bands at 453 cm^−1^ and 1070 cm^−1^, corresponding to Si-O-Si bonds, and at 802 cm^−1^, related to silanol groups (Si-OH) on the surface. These silanol groups may interact with the carbonyl groups of acetone molecules, the VOC model used in this work, potentially impacting catalytic performance [21]. Bulk MnO_x_ exhibits broad bands between 350 and 600 cm^−1^, with peak absorption around 480 cm^−1^ [32]. However, the specific phase of Mn_2_O_3_ (α or γ) cannot be determined from this analysis due to the similarity of FT-IR patterns reported in the literature [32]. Despite this limitation, the results align with the XRD findings. For the Mn40RHS and Mn30RHS catalysts, two characteristic peaks of Mn_3_O_4_ are identified at 475 cm^−1^ and 594 cm^−1^ [32]. In contrast, the FT-IR spectrum of Mn10RHS is similar to that of RHS.

The results of SEM analysis and EDS mapping are presented in Figure 5. The SEM image of MnO_x_ (Figure 5i,j) reveals particle agglomeration, in contrast to the catalysts Mn10RHS (Figure 5c,d), Mn30RHS (Figure 5e,f), and Mn40RHS (Figure 5g,h). EDS mapping was conducted to evaluate the distribution of manganese oxide on the RHS surface, showing no agglomeration zones for the manganese phase, which indicates a homogeneous distribution.

The H_2_-TPR results depicted in Figure 6 illustrate the reduction behavior of the catalyst components, including the support and the catalytic phase. For MnO_x_, a two-step reduction is observed, with peaks at 450 °C and 510 °C. In contrast, the reduction of Mn40RHS occurs in a single step, peaking at 445 °C, while RHS shows no reduction. The total H_2_ consumed (O/Mn) is 0.46 for MnO_x_, indicating that the starting oxide was Mn_2_O_3_, which was subsequently reduced to Mn_3_O_4_ and MnO. The Mn40RHS catalyst shows an H_2_ consumption corresponding to a 0.30 O/Mn ratio, suggesting that the starting oxide is Mn_3_O_4_, consistent with its single-peak reduction profile. This increase in the oxidation state of manganese with its content aligns with the literature [35]. Additionally, the more intense low-temperature reduction peak for Mn40RHS can be attributed to a higher dispersion of the manganese phase on the support.

### 3.2. Activity Testing

The *light-off* curves for the catalysts using acetone as a VOC model are presented in Figure 7. All catalysts and the support demonstrate activity in the catalytic conversion of acetone. For the catalysts containing MnO_x_, acetone oxidation is known to occur via a Mars–van Krevelen mechanism through which the reaction occurs between the adsorbed VOCs and the lattice oxygen of the catalyst rather than the oxygen in the gas phase [36,37].

For RHS, catalytic activity begins above 250 °C, reaching conversion values of up to 90% at 450 °C. This conversion can be partly attributed to the presence of silanol groups on the RHS surface [21]. Bulk MnO_x_ achieves over 90% acetone conversion at 250 °C, while the supported catalysts—Mn10RHS, Mn30RHS, and Mn40RHS—achieve this performance at 300 °C. Although RHS shows activity above 250 °C, it does not reach 10% acetone conversion at 300 °C, indicating its contribution to acetone oxidation is smaller but not negligible, supporting its role as a catalytic support.

Moreover, the *light-off* curves demonstrate the effectiveness of supporting the active phase on RHS at contents below 50%. Under the specified temperature conditions, Mn30RHS and Mn40RHS achieve complete conversion to CO_2_. This behavior is summarized in Table 2, which presents the ratio (R) between experimentally quantified CO_2_ and CO_2_ calculated from the total combustion of acetone, based on experimental conversion at various temperatures. The ratios were evaluated at T_85_ and T_95_, the temperatures at which 85% and 95% acetone conversion are reached, respectively, along with the temperature at which a ratio of 1 indicates total conversion to CO_2_.

While the Mn10RHS catalyst achieved conversions above 90%, it did not attain complete conversion of acetone to CO_2_. In contrast, the Mn30RHS and Mn40RHS catalysts demonstrated significantly better performance, reaching an R value of 1 at 300 °C. This improvement is likely due to better surface coverage of the support in Mn30RHS and Mn40RHS, attributed to higher loading compared to Mn10RHS, despite all catalysts displaying similar phases according to XRD.

Among the Mn30RHS and Mn40RHS catalysts, Mn30RHS exhibited the lowest T_85_ and T_95_ temperatures (see Table 2), indicating superior performance in acetone oxidation. This enhanced activity may stem from its higher surface area, larger pore and mesopore volumes, and improved distribution, as supported by N_2_ physisorption and SEM results. Thus, Mn30RHS is more effective for acetone conversion, allowing for a smaller amount of active phase when using SiO_2_ derived from RH.

The positive results for acetone conversion align with previous research, as shown in Table 3. The T_90_ values obtained in this study are intermediate compared to those reported for manganese oxides on various supports in the catalytic combustion of acetone. It is worth noting that catalysts achieving better T_90_ values than Mn30RHS either incorporate a second metal in the active phase or lack a support.

With Mn30RHS identified as the best supported catalyst for acetone, achieving conversions above 90%, its catalytic stability was evaluated over 24 h under the same flow conditions used for catalytic testing but at 300 °C, where an R value of 1 was achieved. The results are presented in Figure 8.

In this case, the stability tests yielded remarkable results, as no deactivation was observed, and the conversion remained virtually constant within the range of (95.6 ± 2.5)%. Additionally, the R ratio was closely monitored, and its value also remained steady within the range of (0.9 ± 0.1).

## 4. Conclusions

This work involved the preparation, characterization, and evaluation of catalysts with varying loads of MnO_x_ supported on mesoporous SiO_2_ derived from Uruguayan rice husk ashes. The silica from RH ash proved to be a promising catalytic support, achieving over 90% acetone conversion at 450 °C, with improved textural properties compared to previous results.

All catalysts demonstrated activity in acetone oxidation, with Mn30RHS and Mn40RHS showing notable conversion to CO_2_. Among the catalysts examined, Mn30RHS yielded the best overall performance. Notably, comparable catalytic behavior to bulk MnO_x_ was achieved at 300 °C, allowing for a reduced quantity of transition metal oxide when supported on RHS.

This study highlights that SiO_2_ derived from Uruguayan rice husk ashes can effectively contribute to the development of mesoporous solid–MnO_x_ catalysts for VOC removal, offering potential solutions for ash deposition and improving air quality.

## Data Availability

The original contributions presented in the study are included in the article. Further inquiries can be directed to the corresponding authors.

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
