# Peer review of "Catalytic Oxidation of Acetone over MnOx-SiO2 Catalysts: An Effective Approach to Valorize Rice Husk Waste"

_materials, 2024, doi:10.3390/ma17246069_

Round 1

Reviewer 1 Report

Comments and Suggestions for Authors

This work has synthesized SiO2 from rice husk ash and used to catalyze acetone, and the study fits to the journal scope. The finding and results are significant. I suggested it to be considered for publications after minor revisions, specific comments are as below:
1. Why chose rice husk as the raw material? If the biomass is changed? The method is still appropriate?
2. In section 2.2, a specific figure for the synthesis method is expected to be shown.

3. The specific mechanism is expected to be concluded and shown in figure.

Reviewer 2 Report

Comments and Suggestions for Authors

Paper requires major revisions

Catalytic oxidation of acetone over MnOx-SiO2 catalysts: an effective approach to valorize rice husk waste

The findings have the potential to advance biomass valorization, but several aspects need clarification and improvement before the manuscript can be considered for publication.

-        The manuscript requires language polishing to address grammatical errors and typos for improved clarity and readability. 

-        Ensure consistent use of units throughout the manuscript (e.g., Line 176, Page 5). 

-        References should follow the journal's formatting guidelines as outlined in the (Guide for Authors)

-        Lines 300–302 (Page 10): The superior performance of Mn30RHS and Mn40RHS is likely influenced by both better surface coverage and enhanced crystallinity, as indicated by the sharper XRD peaks. This should be discussed alongside BET surface area results for a more comprehensive explanation. 

-        Consider adding a comparative table with results from similar studies on catalytic acetone oxidation to highlight the relevance and effectiveness of your catalysts. 

Comments on the Quality of English Language

The manuscript requires language polishing to address grammatical errors and typos for improved clarity and readability. 

Reviewer 3 Report

Comments and Suggestions for Authors

The manuscript entitled “Catalytic oxidation of acetone over MnOx-SiO2 catalysts: an effective approach to valorize rice husk waste” submitted for publication in the Journal Materials, represents an original study on the synthesis, characterization and catalytic application of rice husk silica, derived from Uruguayan rice husk, as support for MnOx catalysts. The effectiveness and stability of the catalyst MnOx/SiO2 with three different contents of the active phase were investigated in the reaction of acetone oxidation. The reported results will be of interest and possess a potential for practical application in air purification from VOCs and rice husk valorization. The text is well structured and written.

However, the manuscript needs some minor revisions before it is ready for publication.

1. In the described methods 2.3.3. is for Microwave induced plasma, while in the Results section it is stated that ICP was used to determine the silica content (line 195).

2. It is good to note in the description of the methods that when determining 2.3.4. Textural characteristics by N2 physisorption, the Non local Density Functional Theory (NLDFT) method was applied to determine the pore size distribution.

3. In 2.3.6. EDS is Energy Dispersive (not Dispersion) Spectroscopy.

4. In Fig. 3 it would be more appropriate if the nitrogen adsorption-desorption isotherm of the pure MnOx catalyst is colored to distinguish it from the black line of the x-axis.

5. Why in the EDS mapping of the samples (Fig. 5), the image in (h), corresponding to the elemental distribution for the Mn40RHS sample, looks like more intensive that the image in (j), corresponding to the pure MnOx?

Round 2

Reviewer 2 Report

Comments and Suggestions for Authors

See overall appreciation in report 2